# State of the Art in the Standardization of Stromal Vascular Fraction Processing

**DOI:** 10.3390/biom15020199

**Published:** 2025-01-30

**Authors:** Martina Cremona, Matteo Gallazzi, Giulio Rusconi, Luca Mariotta, Mauro Gola, Gianni Soldati

**Affiliations:** 1Swiss Stem Cell Foundation, 6900 Lugano, Switzerland; martina.cremona@sscf.ch (M.C.);; 2Swiss Stem Cells Biotech AG, 8008 Zürich, Switzerland

**Keywords:** adipose tissue, Stromal Vascular Fraction, GMP, adipose tissue processing, cryopreservation, quality controls

## Abstract

Stromal Vascular Fraction (SVF) has gained significant attention in clinical applications due to its regenerative and anti-inflammatory properties. Initially identified decades ago, SVF is derived from adipose tissue and has been increasingly utilized in a variety of therapeutic settings. The isolation and processing protocols for SVF have evolved substantially, particularly after its classification as an Advanced Therapy Medicinal Product (ATMP), which mandates adherence to Good Manufacturing Practices to ensure sterility and product quality. Despite the progress, few studies over the last decade have focused on the standardization of SVF processing. Recent advances, driven by the potential of SVF and its derived products such as Adipose-derived Stem Cells, have prompted the development of improved isolation strategies aimed at enhancing their therapeutic and regenerative efficacy. Notable progress includes the advent of automated processing systems, which reduce technical errors, minimize variability, and improve reproducibility across laboratories. These developments, along with the establishment of more precise protocols and guidelines, have enhanced the consistency and clinical applicability of SVF-based therapies. This review discusses the key aspects of SVF isolation and processing, highlighting the efforts to standardize the procedure and ensure the reliability of SVF products for clinical use.

## 1. Introduction

Stromal Vascular Fraction (SVF) is composed of a heterogeneous population of cells, such as adipose-derived stem cells (ASCs), pericytes, and smooth muscle cells [1,2]. Since its first identification in the 60s as a fraction of non-adipocytes cells [3], SVF has been widely characterized and consequently employed in different clinicasl settings mainly due to regenerative and anti-inflammatory properties [4]. The enzymatic isolation of SVF derived from adipose tissue (AT) was first performed in 2006 by Yoshimura and colleagues through tissue incubation with a collagenase solution at 37 °C, followed by several centrifugation and washing steps which allowed the gradient separation of digested tissue in aqueous phase, enriched in SVF cellular components, from waste oil phase [5]. The enzymatic isolation protocol has been greatly improved and standardized over decades, because of the classification of SVF as an Advanced Therapy Medicinal Product (ATMP) due to the process with substantial manipulation (enzymatic digestion). Once considered as ATMP, SVF must be isolated according to good manufacturing practices (GMP) to ensure the highest sterility and quality of final products during the process. Moreover, the percentage of cell viability and the total number of viable cells (VNCs) per mL detected in each AT were considered as main specifications of the final product, as they reflect the extraction efficiency as well as the quality of the processed sample, anticipating the optimal starting point to trigger the best potential therapeutic effect. Both cell viability percentage and VNCs were demonstrated to be influenced by AT harvesting techniques and different isolation protocols performed through different approaches, i.e., manual, automated, enzymatic, or mechanic [6,7,8,9]. The multicolor flow cytometry analysis was proven to be a powerful technique to discriminate all different cell subpopulations present in the SVF, after the design of specific gating strategies to estimate the distribution of nucleated cells [10,11]. Few works on SVF process standardization have been published in the last 10 years [12,13]; however, the increased sensitivity of ATMPs to be used as drugs and new SVF/ASCs-derived candidate drugs, triggered researchers to develop new strategies to improve the therapeutical and regenerative potential of SVF and SVF-derived products (https://clinicaltrials.gov/, accessed on 9 November 2024) [14]. Indeed, SVF has gained significant interest in regenerative medicine due to the anti-apoptotic, antioxidant, anti-inflammatory, and pro-angiogenic properties of mesenchymal components (ASCs) [2,15,16,17]. These features have led to its therapeutic application in various clinical settings. Recent studies reviewed by Goncharov and colleagues [4] highlight the broad potential of SVF in clinical medicine. Notably, SVF has been used in orthopedics to treat knee osteoarthritis (OA), resulting in improved pain management, cartilage regeneration, and mobility [18,19,20]. In reconstructive surgery, SVF-based therapy has been applied in nasal repair and breast reconstruction, offering improved aesthetic outcomes, and expanding treatment options for critical areas [21,22,23]. SVF has also shown promise in early-phase clinical trials for treating myocardial ischemia [24,25]. Additionally, a systematic review by Ferreira et al. [26] revealed that SVF therapy spans a wide range of applications, from cosmetic and aesthetic uses to clinical treatments for neurological and neurodegenerative diseases (e.g., Parkinson’s and Alzheimer’s), autoimmune diseases, infertility, and vocal cord injuries [27,28]. The growing body of evidence on the efficacy and safety of SVF therapy encourages further studies to explore its broader applications. Future research should focus on the reproducibility of results and the development of standardized protocols for isolating SVF.

With this aim, our group published, in a collaborative work, a proposal strategy for the standardization of GMP-compliant isolation and analysis processes, performed and validated at two different locations [11]. Standardization of SVF processing still retains several challenges due to physiological variabilities (i.e., patient-specific factors such as age, sex, body mass index, health status), SVF cellular heterogeneity, process variabilities, and the lack of regulatory standards [18,29].

Despite these challenges, several advances have been achieved in the standardization of SVF processing, whereas the development of automated processing systems allowed the reduction of both technical errors and variability in SVF isolation, as well as the matching of several protocols, implying the publication of precise guidelines to minimize variability and enhance reproducibility of the SVF processing across different laboratories. Moreover, crucial advances in flow cytometry and molecular techniques for immunophenotyping allowed an increased resolution to precisely discriminate different cell subpopulations [30,31]. Still, different surgical approaches and facility processing of AT induced several pending questions about the standardization more than two decades after the paper by Zuk in 2001 [2]. Here we discuss different aspects trying to figure out all common denominators of the whole process.

## 2. Methods

### 2.1. Research Strategy

Extensive literature research was conducted using PubMed, from 1987 to November 2024. The 30-year time frame gave a “historical” background of adipose tissue processing from the first extraction to the latest methods in use. The keywords used were “stromal vascular fraction”, “stromal vascular fraction isolation protocols”, “adipose tissue harvesting techniques”, “adipose tissue enzymatic digestion”, “adipose tissue mechanical digestion”, “svf cryopreservation” and “process quality controls”.

### 2.2. Selection Criteria

Inclusion criteria comprised both reviews and research articles. Selected studies included prospective and/or retrospective case series, and review articles that explored the harvesting, transport, and processing of adipose tissue under GMP conditions, both enzymatic and non-enzymatic SVF isolation, SVF cryopreservation, processing quality controls as well as therapeutic applications of SVF. Studies that were not peer-reviewed, were excluded. Likewise, articles not available in English, letters to the editor, studies that did not provide explicit data on SVF, and duplicate studies were also omitted from this review. For clinical trials, only those that reported clear methodologies, patient outcomes, and statistical analyses were considered. Articles had to be explicitly focused on adipose tissue processing and SVF isolation under GMP conditions.

## 3. Tissue Harvesting and Transport

Usually AT samples are collected from patients undergoing liposuction procedures and performed by accredited plastic surgeons. These procedures are typically conducted under either local or general anesthesia, and approximately 150 mL of AT are collected. Both manual and automated liposuction techniques are utilized to extract the tissue. Figure 1 shows as an example, the variability of volume collection by different surgeons on N = 297 patients processed in our facility.

In the manual technique, liposuction is performed using cannulas of varying sizes, such as 25 cm and 50 cm lengths, with diameters of 4 mm and 2 mm aspiration holes, respectively. These cannulas allow for precise manual fat removal. On the other hand, the Power-Assisted Liposuction (PAL™, Charlottesville, VA, USA) technique, developed by Dr. Malak in the late 1990s [32], is the most used automated method. The PAL system uses soft, reciprocating cannulas that glide more easily through tissue, significantly reducing the physical effort required by the surgeon. This results in a more efficient procedure, shortening the surgery time and improving patient comfort. Once AT is collected, it undergoes a processing step to separate the aqueous and oily phases. The tissue is placed in sterile syringes and held vertically to allow the phase separation, which helps eliminate the lower, non-cellular phase enriched with blood residues and Klein’s solution (a tumescent solution composed of 500 mL saline solution, 10 mL 2% lidocaine, sodium bicarbonate, and epinephrine). The upper phase, which contains the viable AT, is carefully extracted for further processing. While manual and PAL liposuction techniques are frequently employed, other promising approaches have been developed to improve the efficiency and precision of fat extraction. One of the most used techniques is tumescent liposuction, first described by Dr. Jeffrey Klein in the 1990s [33,34]. Tumescent liposuction involves the local injection of a mixture of saline, anesthetic, and epinephrine into the fat, which reduces bleeding and facilitates the removal of AT. Another technique, Ultrasound-Assisted Liposuction (UAL), uses ultrasound energy to liquefy fat tissue before it is removed, allowing for more precise and less traumatic liposuction [35]. Further advancements in liposuction technologies include Laser-Assisted Liposuction (LAL)1 and Water-Assisted Liposuction (WAL). In LAL, laser energy is used to liquefy the fat cells, enabling easier removal. This method has the added benefit of promoting skin tightening through collagen stimulation. The “SmartLipo” system, a well-known LAL device, liquefies fat cells through heat and facilitates easier fat removal via a specialized cannula. Additionally, WAL uses a pulsating water jet to disaggregate the fat cells, resulting in less trauma to the tissue and faster recovery times for patients [36,37]. Both LAL and WAL have been shown to reduce procedure time, enhance recovery, and improve aesthetic outcomes. Each liposuction technique has its own indications, benefits, and potential risks. The selection of the most appropriate method is typically based on factors such as the specific goals of the surgery, the characteristics of the patient (e.g., body area to be treated, skin quality), and the surgeon’s expertise. Although many different collection techniques have been developed and used by many different surgeons, we didn’t notice any statistical difference between surgeons in terms of the total nucleated cells collected by various collection methods as shown in Figure 2.

Transportation of samples to a GMP-compliant laboratory is a key aspect of ensuring the highest quality and integrity of the tissue for subsequent processing and use in regenerative medicine or research. Unfortunately, the importance of proper sample transportation has often been overlooked in the scientific community, despite its significant impact on the quality of the samples. To comply with GMP regulations, tissue samples should be transported within 24 h after the harvesting in sterile bags within temperature-controlled boxes, maintained at room temperature (RT) or preferably, at 4 °C, and equipped with temperature-monitoring probes to ensure that the proper storage conditions are maintained throughout the transport process [38,39]. This guarantees that the tissue remains viable upon arrival at the laboratory for further processing. Once samples are delivered, they should be processed as soon as possible to minimize any degradation or loss of cell viability. Proper tissue handling, transportation, and processing are equally critical to ensure high-quality samples for downstream applications. By refining these protocols and adhering to GMP standards, it is possible to enhance the reproducibility and success of studies and therapies based on adipose-derived cells [40,41,42].

## 4. SVF Isolation

SVF isolation represents a critical step in extracting a viable population of ASCs and stromal cells for regenerative purposes. This process follows the disruption of AT using enzymatic or mechanical dissociation methods to obtain a heterogeneous population of cells, including ASCs, pericytes, fibroblasts, immune cells, and other progenitor cells [43,44,45]. Enzymatic digestion has long been considered the “gold standard” for SVF isolation due to its ability to effectively dissociate AT and release a high yield of viable cells [46]. This method typically involves the use of collagenase, an enzyme that breaks down collagen of the extracellular matrix, allowing for the separation of stromal cells from the AT. A commonly used collagenases blend consists of type I and VIII collagenases, resuspended in enzyme-activating phosphate-buffered saline (PBS) containing calcium and magnesium ions (Ca2+/Mg2+) [30]. The tissue is incubated at 37 °C, followed by serial centrifugation and washing steps using PBS without Ca2+/Mg2+ to block enzyme activity and optimize cell viability [47,48]. Typical enzymatic protocols for SVF isolation involve collagenase I, III, and V, which are incubated for 40–60 min at 37 °C under constant and gentle agitation. This is followed by multiple filtration steps through 100 µm and 40 µm cell strainers to remove undigested tissue, cell debris, and extracellular matrix components. These protocols ensure high cell yields, with viability exceeding 70% and high colony-forming unit (CFU) values, which are indicative of functional ASC populations [39,49]. The application of enzymatic digestion protocols in GMP-compliant settings presents several challenges. Since the process is considered “more than minimal manipulation” the product classification as an ATMP complicates its translation into clinical application because compliance with GMP standards is required. To address these challenges, a variety of automated and non-enzymatic isolation devices have been developed that utilize physical forces (e.g., mechanical separation or centrifugation) have been developed to extract stromal cells. These devices are particularly valuable in intraoperative settings, where rapid processing of AT for autologous grafting is preferred. Several automated systems use collagenase-based digestion for SVF isolation, including the Celution^®^ 800-CRS and 820-CRS (Cytori Therapeutics, San Diego, CA, USA), STEM-X™ (Medikan International, Heojun-ro, Gangseo-gu, Seoul, Republic of Korea), and Sepax 2 (Biosafe-Group, Eysins, Switzerland) [8,50,51] These systems standardize the SVF isolation process and enhance reproducibility, making them ideal for use in regenerative medicine, where consistency and sterility are paramount. Furthermore, these devices are capable of isolating and enriching AT for intraoperative autologous grafting, improving surgical outcomes. Other innovative systems include the Beauty Cell (N-Biotek, Pyeongcheon-ro, Bucheon-si, Gyeonggi-do, Korea), AdiStem™ Kit (AdiStem Pty Ltd., Melbourne, Australia), and Sceldis^®^ (Purebiotech, Seongnam-city, Kyunggi, South Korea), each designed to streamline the SVF isolation process while maintaining high cell viability [8,50]. These devices aim to facilitate the translation of AT-derived therapies into clinical practice by reducing the complexity of manual processing. Mechanical isolation techniques provide an alternative to enzymatic digestion, offering an approach that preserves cell surface integrity. These methods involve dissociating AT using mechanical forces such as blades of varying sizes or steel beads to break up the tissue and release the SVF [52,53]. After mechanical dissociation, the tissue undergoes serial filtration and centrifugation steps [1,54]. The advantage of mechanical methods is that they can be performed without enzymes by reducing the process to a “minimal manipulation” from a regulatory point of view. Mechanical methods such as Lipogems^®^ yield tissue derived SVF (tSVF), which still contains a residual extracellular matrix, offering an additional scaffold that can be beneficial for regenerative applications. The extracellular matrix in tSVF serves as a structural framework for the cells and a reservoir for paracrine factors, which can enhance tissue regeneration in vitro and in vivo [55,56,57,58]. While mechanical digestion typically results in a lower total cell yield compared to enzymatic methods, studies have shown that SVF-derived cells obtained through mechanical dissociation maintain similar morphological and functional characteristics as those isolated using enzymatic protocols [59]. To further improve SVF isolation, hybrid methods that combine enzymatic digestion with mechanical dissociation have been explored. For example, the combination of Liberase™ Blendzyme with ultrasound for cell membrane permeabilization has been shown to enhance both cell yield and the overall efficiency of the process [60]. These combined approaches leverage the advantages of both methods, facilitating the efficient disruption of AT while minimizing the degradation of cellular components. The choice of method for AT dissociation depends on several factors, including the volume of AT to be processed, the safety and reproducibility of the protocol, and the specific clinical application of the isolated cells. For instance, when large volumes of AT are required for regenerative procedures, enzymatic digestion may be preferred due to its higher cell yield and the possibility of concentrating the cells. Conversely, for intra-operatory applications mechanical methods or hybrid techniques may offer a better alternative [54,59]. The evaluation of total cell yield, as well as cell viability, might be another selection criterium even though, following a recent systematic review by Uguten and colleagues, both mechanical and enzymatic isolation procedures showed comparable results as shown in Figure 3. Specifically, 21 different isolation protocols, both enzymatic and mechanical have been compared with a cell yield ranging from 2.3 × 10^5^–18 × 10^5^ for enzymatic process and 0.3 × 10^4^–26.7 × 10^5^ cells/mL for mechanical processes. Viability ranged from 70–99% in enzymatic procedures and was slightly lower in mechanical processes ranging from 46–97.5% [61]. In conclusion, both enzymatic and mechanical methods have their strengths, and the selection of the optimal technique depends on the specific needs of the procedure. Advances in automated devices and hybrid techniques are making SVF isolation more efficient, reproducible, and clinically feasible, paving the way for more widespread use of adipose-derived cells in regenerative medicine.

## 5. SVF Cryopreservation

Cryopreservation of SVF samples is typically performed using either liquid nitrogen or vapor-phase nitrogen. However, the optimal formulation of the freezing medium remains a subject of debate, primarily due to the lack of a standardized, universally accepted protocol. The methods for cryopreservation can generally be categorized into slow-freezing, vitrification, low-temperature subzero preservation, and dry-state preservation [62]. Each of these methods differs primarily in the concentration and type of cryoprotective agents (CPAs) used, as well as the cooling rates, which are crucial factors for minimizing cell damage during freezing. The ideal cooling rate for cryopreservation can vary depending on the cell type, as different cell types exhibit varying rates of water transport across their plasma membranes. During slow freezing, for example, water in the cytoplasm is gradually replaced by CPAs, which helps to reduce cellular damage associated with ice crystal formation. Cooling rates are carefully controlled to match the permeability of the cell membrane to the CPA. Slow cooling protocols, particularly when using controlled-rate freezers or portable benchtop freezing containers, typically involve cooling rates of around 1 °C per minute with CPA concentrations lower than 1.0 M. This slow cooling process is often preferred for cryopreserving mesenchymal stem cells (MSCs) due to the relatively simple handling of samples, making it well-suited for clinical and research settings [63]. Recent advances have explored methods to enhance the efficacy of cryopreservation. For instance, Zifei Li and colleagues have demonstrated that the addition of cryoprotective agents combined with nanographene oxide (nano-GO) can significantly improve the preservation of AT and ASCs during both the freezing process and long-term cryopreservation [64]. Similarly, another study pointed out that the presence of hydrogen gas during the freezing process enhances cell viability and restores biological functions after thawing [65]. These novel approaches suggest that additional factors, such as nanomaterials or gases, can mitigate cryoinjury and improve overall cell survival. In parallel, efforts have been made to develop xeno-free cryopreservation media suitable for the storage of stromal cells isolated from AT, improving the safety and efficacy of fat grafting procedures. Several studies have investigated cryopreservation media formulations that utilize Dimethyl Sulfoxide (DMSO) in combination with serum albumins—either Bovine Serum Albumin (BSA) or Human Serum Albumin (HSA)—to preserve cell viability, differentiation potential, and proliferative capacity. Research has demonstrated that such media help maintain high growth rates for SVF cells and ASCs, while also minimizing cell loss after thawing [66,67,68]. For clinical applications, cryopreservation of SVF and ASCs for patient use must adhere to stringent GMP-compliant protocols to ensure the quality and safety of the products. In a pre-clinical study, a cryopreservation protocol using a medium containing 5% human albumin and 5% DMSO demonstrated high viability and long-term storage capacity, with cells remaining viable even after extended periods, up to at least 10 years [69]. These findings underline the importance of optimizing cryopreservation protocols to preserve the functionality and therapeutic potential of SVF and ASCs for future clinical applications, including regenerative medicine and fat grafting. Moreover, significant advantages of cryopreserving SVF over fat and skin grafts must be highlighted, in terms of regenerative potential, storage flexibility, lower risk of complications, improved graft survival, and long-term clinical applications. The ability to store SVF and use it on-demand also provides a level of flexibility and planning that fat and skin grafts cannot match, making it a promising option for a wide range of clinical and aesthetic purposes. SVF thawing is another crucial aspect to maximally reduce osmotic shock, and eventual ice recrystallization, and maintain a high cell count and viability [70]. In conclusion, while cryopreservation of SVF and ASCs holds great promise for clinical applications, continued refinement of freezing techniques, CPA formulations, and storage conditions is critical. Advances in novel cryoprotectants, xeno-free media, and alternative preservation strategies, combined with rigorous clinical protocols, will be essential for enhancing the long-term viability and functionality of these valuable cellular products.

## 6. Quality Controls: A Brief Introduction

Quality control (QC) is an important procedure in cell-based therapies. Ensuring the sterility, viability, and functionality of SVF cells, QC is paramount to the success of clinical applications. Microbiological contamination during the GMP-compliant processing of AT represents a significant risk, which must be rigorously controlled. Microbiological contaminations are mitigated through environmental monitoring and microbiological analyses, conducted in accordance with European Pharmacopoeia [71] guidelines, Eudralex Annex 1, and other GMP standards. Moreover, a periodic quality review and the root cause analysis of any OOS (out of specification) results improve the overall quality of final products. Despite the importance of these protocols, there remains a lack of consensus regarding standardized QC procedures specifically tailored to SVF processing.

### 6.1. Microbiological Testing and Environmental Control

According to European Pharmacopoeia [71], microbiological contamination in SVF preparations can be assessed by inoculating small volumes of the extracted SVF into Bact/Alert culture bottles, both aerobic and anaerobic, which contain 40 mL of culture medium. These culture bottles are incubated at 37 °C with 5% CO_2_ for 10–14 days and continuously monitored every 10 min using the Bact/Alert system [72,73]. This automated system can detect microbial growth early, ensuring timely identification of contamination [71]. Moreover, bacterial endotoxins, primarily produced by Gram-negative bacteria, are another significant concern in microbiological testing. Endotoxins are toxic to humans and can cause severe inflammatory reactions. Endotoxin testing is typically performed using the Limulus Amebocyte Lysate (LAL)2 test, which involves the use of LAL derived from the blood of the horseshoe crab (Limulus polyphemus) [71]. The LAL test utilizes gel-clot, turbidimetric, or chromogenic techniques, each offering different sensitivities and advantages depending on the application [71]. An alternative to the traditional LAL test is the PyroSense kit, which uses recombinant Factor C (rFC). rFC is a recombinant endotoxin-activated enzyme, offering the potential for a more reproducible and scalable assay. Several studies, such as those by Ding and Ho [74] and more recent comparisons by Dubczak and colleagues [75] have shown that while rFC assays are efficient, they may underestimate endotoxin activity relative to the LAL test, highlighting the need for careful selection of the appropriate test for endotoxin detection in GMP environments. This is particularly relevant given the increasing use of recombinant products in regenerative medicine [74,75,76].

Besides microbiological testing, environmental monitoring is another crucial aspect for ensuring the sterility of the processed SVF product. Eudralex Annex 1 guidelines for the manufacture of sterile medicinal products [76] outline specific requirements for environmental control during the GMP process, including air quality and personnel hygiene. Effective environmental monitoring ensures that the final product meets the stringent sterility requirements mandated for GMP compliance. Environmental testing methods include the use of sedimentation plates for continuous microbiological control, as well as microbiological active sampling plates for air monitoring. According to the PIC/S guidelines [77], active air sampling is recommended for areas with a controlled atmosphere (such as cleanrooms) to assess the microbial load in the air, ensuring that the air quality meets the required standards. Contact plates are also used at the end of the process, to assess the cleanliness of surfaces that may come into direct contact with the SVF sample ([76], paragraph 9.30; [77] annex 1, paragraph 4.31). Monitoring such surfaces is crucial as contamination can affect product sterility and, subsequently, patient safety. Continuous particle monitoring is another key aspect of environmental control as eventual particle contamination can introduce microbial vectors or contribute to the physical degradation of cells, thus compromising the therapeutic product’s quality ([76], paragraph 4.27; [11,77], annex 1, paragraph 4.27). Despite existing guidelines, there are several challenges to implementing standardized QC protocols for SVF isolation which lie in the variability of SVF extraction methods, which can affect the overall quality of the final product.

As noted by François et al. (2021) [11], there is currently no consensus on a standardized protocol for SVF isolation, leading to inconsistencies in QC practices across different laboratories and GMP facilities. Additionally, the complexity of AT itself, with its heterogeneous composition of cells, extracellular matrix, and adipocytes further complicates QC measures, as all components need to be carefully processed and monitored. One promising area for improvement is the development of more automated and integrated systems combining both environmental and microbiological monitoring in real-time during SVF processing. Such systems could help to streamline the QC process and reduce human error, enhancing both the safety and reproducibility of the final product. The integration of advanced technologies like next-generation sequencing (NGS) and real-time PCR for microbial detection could offer higher sensitivity compared to traditional methods and facilitate the identification of microbial contaminants at lower concentrations [78].

Effective QC for SVF isolation requires a multifaceted approach, combining rigorous microbiological testing with comprehensive environmental monitoring and standardized characterization of SVF-derived cells. While current guidelines provide a robust framework for ensuring sterility and product safety, challenges remain in establishing universally accepted protocols for SVF isolation and quality assessment. Moving forward, advancements in automated systems, integrated monitoring techniques, and improved flow cytometry protocols will be essential in ensuring the reproducibility and safety of adipose-derived cell-based therapies.

### 6.2. Flow Cytometry Analysis

Another area of potential improvement is the development of better-standardized flow cytometry protocols to assess the quality and identity of SVF-derived cells. A more universally accepted gating strategy for SVF characterization would allow for a more consistent evaluation of subpopulations, ensuring the uniformity of the isolated cells, which is crucial for their clinical efficacy [11]. Flow cytometry analysis is essential for quality assurance, as it allows for the assessment of cell viability, phenotype, and potential contamination, thus serving as a critical QC tool for SVF-based therapies.

The SVF is a complex, heterogeneous mixture of cells, comprising a variety of cell types, including MSCs, endothelial cells, hematopoietic cells, pericytes, and others [79,80]. To accurately identify these cell subsets within the SVF, multicolor flow cytometry is essential, as it provides a high-dimensional analysis that allows for precise discrimination between cell populations. This approach plays a critical role in ensuring the QC of cellular products, particularly in regenerative medicine applications. Despite its importance, the gating strategy used for SVF analysis remains poorly standardized, limiting its widespread adoption and reliable interpretation [11,79]. The minimal criteria for the characterization of MSCs were first defined in 2006 by the International Society for Cell & Gene Therapy (ISCT), which established a set of surface markers (CD73, CD90, CD105) that should be expressed on MSCs, along with a lack of hematopoietic markers like CD45 [79]. These criteria were revised in 2013 in collaboration with the International Federation for Adipose Therapeutics and Science (IFATS) to better reflect the diverse applications of adipose-derived stem cells [80,81]. However, one major limitation of these criteria is that the commonly used markers for MSC identification—CD73, CD90, and CD105—are not specific to a single cell population and often overlap across various subsets within the SVF. For example, the CD73+CD90+CD105+ phenotype encompasses not only MSCs but also other cell types such as endothelial cells and perivascular cells, posing challenges for precise subpopulation identification in the SVF. In response to these challenges, a collaborative study conducted in 2021 between the Swiss Stem Cell Foundation (SSCF) and the University Hospital of Marseille developed a more refined and precise gating strategy for SVF analysis using multicolor flow cytometry as shown in Figure 4 here below.

This enhanced approach allows for a more accurate sorting and identification of cell subsets in the SVF’s complex mixture. The gating process begins with the elimination of cell debris and aggregates using SYTO 40, a nucleic acid-binding dye that allows the selection of nucleated cells. Following this, viable cells are distinguished using 7-aminoactinomycin D (7-AAD), a dye that marks non-viable cells, ensuring that only viable cells are considered in downstream analyses. Once viable cells are isolated, they are differentiated into two broad categories based on the expression of CD45: hematopoietic (CD45+) and non-hematopoietic (CD45-) cells. The non-hematopoietic population is then further subdivided using CD146 and CD34 markers. The CD34+CD146- subset corresponds primarily to ASCs, while the CD34+CD146+ subset is enriched in endothelial cells. This dual-marker approach significantly improves the specificity of subpopulation identification within the SVF, as CD34 and CD146 markers are more distinctively associated with specific cell types within the non-hematopoietic fraction. This advanced multicolor flow cytometry gating strategy has gained widespread adoption in recent years and has proven to be invaluable for the precise identification of the various subpopulations within the SVF. Studies employing this technique have provided insights into the functional diversity of SVF components, including research into sex-related immunophenotype differences [81] and alterations in cell populations during the early stages of lipedema [82].

In conclusion, while the standardization of SVF gating strategies has been a long-standing challenge, the adoption of this refined multicolor flow cytometry approach represents a significant step forward in the characterization of SVF cell populations. By offering a more accurate and reproducible method for discriminating between cell subsets, this strategy not only enhances our understanding of SVF composition but also contributes to the advancement of cell-based therapies.

### 6.3. Additional QC Tests

Besides microbiological contamination, environmental control, and flow cytometry, several other methods can be employed to assess the quality of isolated SVF. The initial quality indicators of an SVF sample include the total nucleated cell count (TNC) and cell viability. These parameters can be assessed using various techniques, such as manual counting with a hemocytometer, flow cytometry, or automated cell counters like the Nucleocounter [82,83,84]. According to recent guidelines, cell viability greater than 70% is considered acceptable for therapeutic products [80,84].

The CFU-F assay is another functional method used to evaluate the frequency of MSCs within the SVF [80,83]. This procedure was first described by Zuk and colleagues [2], taking advantage of ASC’s property to be plastic adherent.

As described by Bourin and colleagues, SVF cells are cultured at low densities (maximum of 400 cells/cm^2^) to allow individual colonies to separately grow in an appropriate culture medium. The number of colonies formed reflects the frequency of stromal progenitors and enhances the quality of the cell therapy product. Differentiation assays are widely used to confirm the multipotent capacity of MSCs within a given cell therapy product. These assays are performed by exposing MSCs to specific culture conditions that promote differentiation into different lineages, typically adipocytes, osteocytes, or chondrocytes. The differentiation potential is typically assessed through morphological observations, as well as biochemical or molecular markers specific to each lineage. However, both differentiation assays and CFU-F require long incubation periods and therefore cannot be performed for immediate product release. Instead, they should be considered for informational purposes to ensure product quality during the development or research phases.

## 7. Conclusions

The aim of this review stands to overview the state of the art for the standardization of the AT-derived SVF processing following GMP regulations, to better approach possible therapeutic applications, and to contribute to the creation of a common, consensus protocol spanning from the collection of the AT to the clinical application of SVF and derivatives. Several improvements have been achieved in the last few years but, unfortunately, a consensus on a common processing protocol, together with the standardized characterization is still lacking. However, the increasing interest in using SVF as a therapeutic tool could contribute to the development of a standardized protocol, recognized by the main regulatory agencies.

## Figures and Tables

**Figure 1 biomolecules-15-00199-f001:**
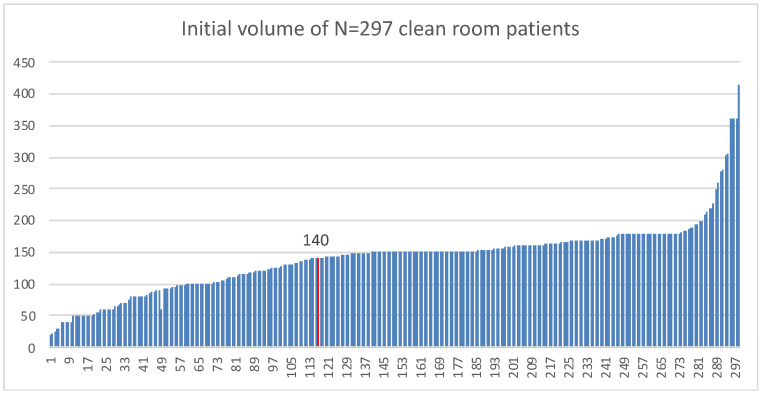
Collection volume data on N = 297 sequential patients. Average volume recorded at 140 mL of adipose tissue collected.

**Figure 2 biomolecules-15-00199-f002:**
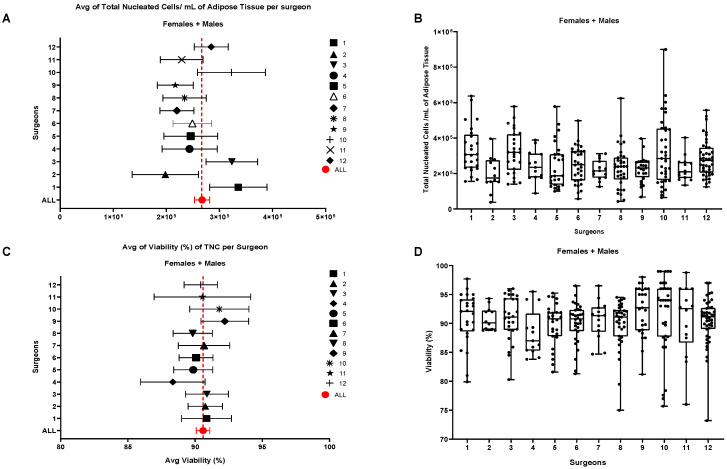
(**A**,**B**). Influence of the Surgeon on the number of Total Nucleated cells (TNCs) per mL of adipose tissue and on the cellular viability. Comparison of the number of TNCs/mL between samples harvested by different Surgeons. (**C**,**D**). Cell viability compared with the total average of the 302 samples. Females + Males cohort N = 302; Females N = 191, Males N = 111 [35].

**Figure 3 biomolecules-15-00199-f003:**
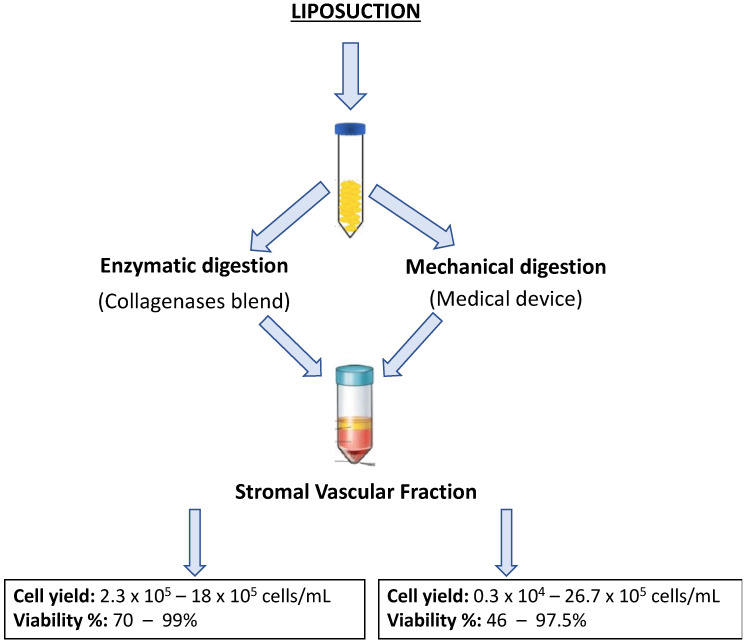
Impact of different SVF isolation approaches. Data shown in the figure are extracted from the following reference [61].

**Figure 4 biomolecules-15-00199-f004:**
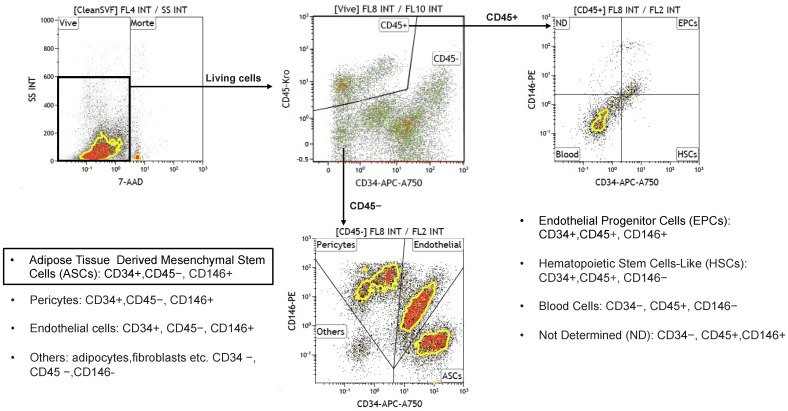
Gating strategy for the characterization of different cell subpopulations within the SVF, extracted from adipose tissue samples through multi-color flow cytometry. CD45+ subpopulations: EPCs, HSCs, Blood cells and ND cells; CD45– subpopulations: ASCs, Endothelial Cells, Pericytes and Others.

## Data Availability

The raw data showed in Figure 1 of this article will be made available by the authors.

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
