# Peer review of "State of the Art in the Standardization of Stromal Vascular Fraction Processing"

_biomolecules, 2025, doi:10.3390/biom15020199_

Round 1

Reviewer 1 Report

Comments and Suggestions for Authors

The article was reviewed.

The concerns were listed as below: 

1.        The introduction section could benefit from additional information on the clinical applications of SVF.

2.        Although the article is of a review nature, it would be helpful to specify the search scope and the years covered in the literature review (if possible).

3.        In the section on cryopreservation, it would be useful to explain if there are any conceptual or practical differences in preserving SVF compared to other tissues (e.g., fat grafts, skin grafts).

4.        Lines 100-101: The composition of the tumescent solution should include sodium bicarbonate.

5.        In the paragraph discussing flow cytometry as a QC process, while the importance of CD markers is emphasized, there is a lack of clarity regarding the standardization of the process of isolating cells from SVF.

6.        The authors' results indicate that the quantity and viability of SVF cells are not affected by the method or site of liposuction but are slightly influenced by gender and age. However, there is no comparative data on how different isolation and cryopreservation steps affect cell quantity and viability.

7.        Is standardized processing correlated with consistent percentages or counts (per ml) of the main component cells in SVF?

Reviewer 2 Report

Comments and Suggestions for Authors

Stromal Vascular Fraction (SVF) is highly valued in clinical applications due to its regenerative and anti-inflammatory properties. This review discusses the key aspects of SVF isolation and processing, emphasizing efforts to standardize procedures and ensure the reliability of SVF products for clinical use. Though the subject is worth considering for publication, several issues need to be addressed.

Comment 1: The author should provide a detailed description of the methods used to collect and analyze data, including search strategies, inclusion and exclusion criteria, and methods for data synthesis.

Comment 2: The author needs to compare and analyze the SVF treatment SOPs used in different studies, discuss the differences between them, and the potential impact of these differences on the results.

Comment 3: Charts and visual aids should be clear, accurate, and helpful in understanding the text content. Suggest adding more charts to compare the effectiveness of different SVF processing techniques.

Comment 4: The review should provide a detailed description of the QC process, including microbiological testing, environmental control, and flow cytometry analysis, and how they affect the quality and safety of SVF products.

Comment 5: As a reviewer, I would suggest the authors to enrich their manuscript by integrating more references from the existing literature to fortify the study's context and enhance its academic rigor. 

Reviewer 3 Report

Comments and Suggestions for Authors

I would suggest two additions to the manuscript. First, I propose to extend the review of the current state of knowledge regarding methods of obtaining adipose tissue and subsequent procedures for the preparation of biologically active SVF preparations, and second, I propose to extend the manuscript with available information regarding the risk of causing fat embolism of vessels as a result of adipose tissue collection. The lack of this second information indicates a lack of real calculation of the risk of the procedure in clinical conditions, especially in relation to aesthetic medicine procedures.

Round 2

Reviewer 1 Report

Comments and Suggestions for Authors

The manuscript has been revised in response to the previously raised questions.